# SPECTRAL NEURAL GRAPH SPARSIFICATION

## ABSTRACT

Graphs are central to modeling complex systems in domains such as social networks, molecular chemistry, and neuroscience. While Graph Neural Networks, particularly Graph Convolutional Networks, have become standard tools for graph learning, they remain constrained by reliance on fixed structures and susceptibility to over-smoothing. We propose the Spectral Preservation Network, a new framework for graph representation learning that generates reduced graphs serving as faithful proxies of the original, enabling downstream tasks such as community detection, influence propagation, and information diffusion at a reduced computational cost. The Spectral Preservation Network introduces two key components: the Joint Graph Evolution layer and the Spectral Concordance loss. The former jointly transforms both the graph topology and the node feature matrix, allowing the structure and attributes to evolve adaptively across layers and overcoming the rigidity of static neighborhood aggregation. The latter regularizes these transformations by enforcing consistency in both the spectral properties of the graph and the feature vectors of the nodes. We evaluate the effectiveness of Spectral Preservation Network on node-level sparsification by analyzing well-established metrics and benchmarking against state-of-the-art methods. The experimental results demonstrate the superior performance and clear advantages of our approach.

## 1 INTRODUCTION

Graphs are the natural language of complex systems, from molecules and transportation networks to social and neural interactions. In recent years, Graph Neural Networks (GNNs) have become the dominant paradigm for learning from such data (Bronstein et al., 2017; Zhou et al., 2020), enabling powerful applications in chemistry (Duvenaud et al., 2015), neuroscience (Zhang et al., 2022), and large-scale network analysis (Hamilton, 2020). Yet, despite their success, standard GNNs suffer from two fundamental limitations. First, they rely on a *fixed graph structure*, which prevents them from adapting connectivity to the task at hand. Second, they quickly run into scalability and expressiveness issues, as message passing tends to oversmooth node representations (Oono & Suzuki, 2020) and becomes inefficient in large, dense graphs.

A natural way to overcome these challenges is to let the model itself *reshape the graph*. Rather than treating the input topology as immutable, one can learn transformations that align structure and features in a task-driven manner, while discarding redundant information. This perspective opens the door to two intertwined objectives: designing neural layers that generate adaptive embeddings by evolving the graph, and introducing principled loss functions that sparsify the topology without breaking its spectral integrity.

In this work, we address both aspects through a new architecture, the **Spectral Preservation Network** (`SpecNet`). Our contributions are twofold:

- **The Joint Graph Evolution layer (`JGE`).** A novel mechanism that reparameterizes the graph Laplacian via bilinear transformations, producing embeddings on dynamically learned topologies rather than static input graphs. This layer mitigates oversmoothing and rigidity, enabling richer structure–feature interactions.

- **The Spectral Concordance loss (`SC`).** A loss that sparsifies the graph at the node level by combining Laplacian alignment, feature-geometry preservation, and a sparsity-inducing trace penalty. This formulation removes uninformative nodes while maintaining global spectral properties and feature consistency.

Together, these components allow `SpecNet` to move beyond static message passing: the graph is no longer a constraint, but a variable optimized during learning. We show that this approach provides a principled and flexible framework for *node-level sparsification*, significantly improving compression efficiency and downstream performance compared to existing heuristic or task-specific methods.

In summary, this paper introduces a new paradigm for graph representation learning: embedding layers that actively reshape structure, coupled with spectral losses that guide sparsification. This synergy equips GNNs with both flexibility and stability, paving the way for scalable, spectrum-driven graph learning.

## 2 SPECTRAL PRESERVATION NETWORK

Spectral Preservation Network (`SpecNet`) is a novel spectral-based neural architecture that jointly learns graph structure and node representations through recursive updates of the graph Laplacian and the node feature space. By operating in the spectral domain and decoupling graph topology from input features, `SpecNet` enables the dynamic synthesis of structurally coherent graphs while preserving global properties and informative node characteristics.

Consider a graph $G = (V, E)$ without self-loops, where $V = \{1, \ldots, n\}$ denotes the set of nodes and $E = \{e_1, \ldots, e_m\}$ the set of edges. The structure of $G$ can be algebraically represented in two equivalent forms: via its adjacency matrix or via its incidence matrix. The definition of the adjacency matrix $A \in \mathbb{R}^{n \times n}$ depends on whether $G$ is directed or undirected. In *directed graphs* each edge $e_k = i_k \to j_k$ represents a directed connection from node $i_k$ to node $j_k$: the adjacency matrix $A$ is defined elementwise as Equation 1. In *undirected graphs* each edge $e_k = \{i_k, j_k\}$ is an unordered pair representing a bidirectional connection between nodes $i_k$ and $j_k$: the corresponding adjacency matrix is given by Equation 2.

$$A_{ij} = \begin{cases} 1 & \text{if } i \to j \in E, \\ 0 & \text{otherwise.} \end{cases} \quad (1) \qquad A_{ij} = A_{ji} = \begin{cases} 1 & \text{if } \{i, j\} \in E, \\ 0 & \text{otherwise.} \end{cases} \quad (2)$$

For undirected graphs, $A$ is symmetric by construction.

The incidence matrix $B \in \{-1, 0, +1\}^{n \times m}$ encodes node-edge relationships based on a chosen orientation for each edge. Its entries are defined as:

$$B_{i,k} = \begin{cases} -1 & \text{if node } i \text{ is the tail of edge } e_k, \\ +1 & \text{if node } i \text{ is the head of edge } e_k, \\ 0 & \text{otherwise.} \end{cases} \quad (3)$$

In directed graphs, each edge $e_k = i_k \to j_k$ has an intrinsic orientation, with $B_{i_k,k} = -1$ and $B_{j_k,k} = +1$. For undirected graphs, an arbitrary but fixed orientation is imposed (e.g., by designating the node with the smaller index as the tail and the larger as the head) before applying the same rule.

Let $X \in \mathbb{R}^{n \times f}$ be the node feature, encoding input features, where each row $X_i$ corresponds to node $i \in V$ and contains an $f$-dimensional attribute vector. This matrix serves as the initial representation of node characteristics. The degree matrix $D \in \mathbb{R}^{n \times n}$ is diagonal, with entries $D_{ii}$ equal to the number of edges incident to node $i$. For directed graphs, $D$ can be decomposed as $D = D^+ + D^-$, where $D^+$ and $D^-$ are diagonal matrices capturing in-degrees and out-degrees, respectively. Specifically, $D_{ii}^+$ counts the number of edges directed toward node $i$, while $D_{ii}^-$ counts those originating from it.

### 2.1 JOINT GRAPH EVOLUTION LAYER

The core of `SpecNet` is the Joint Graph Evolution (`JGE`) layer, a novel architectural component that operates on a pair of input matrices: an adjacency matrix $Q_t \in \mathbb{R}^{r_t \times r_t}$ and a feature matrix $H_t \in \mathbb{R}^{r_t \times p_t}$, both sharing the same number of rows. Here, $t$ denotes the layer index within the network. The transformation produces embeddings as updated matrices $Q_{t+1} \in \mathbb{R}^{r_{t+1} \times r_{t+1}}$ and $H_{t+1} \in \mathbb{R}^{r_{t+1} \times p_{t+1}}$, corresponding to a new node set of size $r_{t+1}$ and a space of $p_{t+1}$ features.

The forward computation of the `JGE` at layer $t$ is defined as:

$$J_{t+1} = \Theta_t \, H_t^\top \, U_t \, Q_t \, V_t \, H_t,$$
$$Q_{t+1} = \sigma_1\big(J_{t+1} \, \Phi_t\big), \tag{4}$$
$$H_{t+1} = \sigma_2\big(J_{t+1} \, \Psi_t\big),$$

where $J_{t+1} \in \mathbb{R}^{p_t \times p_t}$ is an intermediate representation, and $\Theta_t \in \mathbb{R}^{r_{t+1} \times p_t}$, $\Phi_t \in \mathbb{R}^{p_t \times r_{t+1}}$, and $\Psi_t \in \mathbb{R}^{p_t \times p_{t+1}}$ are learnable parameter matrices. The functions $\sigma_1$ and $\sigma_2$ denote elementwise nonlinearities. The matrices $U_t, V_t \in \mathbb{R}^{r_t \times r_t}$ are diagonal normalization matrices defined as follows. Define the row-wise and column-wise absolute sums of $Q_t$:

$$[u_t]_i = \sum_{j=1}^{r_t} |(Q_t)_{ij}|, \qquad [v_t]_j = \sum_{i=1}^{r_t} |(Q_t)_{ij}|. \tag{5}$$

The diagonal entries of $U_t$ and $V_t$ are then given by:

$$[U_t]_{ii} = \begin{cases} 1/\sqrt{[u_t]_i}, & \text{if } [u_t]_i > 0, \\ 0, & \text{otherwise,} \end{cases} \qquad [V_t]_{jj} = \begin{cases} 1/\sqrt{[v_t]_j}, & \text{if } [v_t]_j > 0, \\ 0, & \text{otherwise.} \end{cases} \tag{6}$$

This normalization ensures that the matrix product $U_t \, Q_t \, V_t$ is non-expansive with respect to the Euclidean norm, as discussed in Appendix A. This property contributes to the numerical stability of the architecture. Non-expansiveness acts as an implicit regularizer, preventing the uncontrolled growth of feature magnitudes, an issue that can compromise optimization in deep architectures. Unlike explicit normalization techniques such as batch normalization (Ioffe & Szegedy, 2015) or spectral normalization (Miyato et al., 2018), this approach enforces norm constraints by construction, without introducing additional computational branches. Moreover, it contributes to controlling the Lipschitz constant of the network, which has implications for both generalization and adversarial robustness (Gouk et al., 2021; Pauli et al., 2022; Zühlke & Kudenko, 2025).

Since $Q_t$ and $H_t$ correspond to a graph adjacency matrix and a node feature matrix, respectively, in a new space, the `JGE` can be interpreted as a learnable mechanism for jointly evolving both graph structure and node representations. The output $Q_{t+1}$ represents a transformed graph topology with updated edge weights and a redefined node set, while $H_{t+1}$ encodes node features aligned with this new structure.

A Spectral Preservation Network is constructed by stacking multiple `JGE` layers. The initial inputs are defined as:

$$H_0 = X, \qquad Q_0 = A, \tag{7}$$

where $X \in \mathbb{R}^{n \times f}$ is the node feature matrix and $A \in \mathbb{R}^{n \times n}$ is the initial adjacency matrix. This implies $r_0 = n$ and $p_0 = f$, with the initial normalization matrices given by:

$$[U_0]_{ii} = \begin{cases} 1/\sqrt{D_{ii}^-}, & \text{if } D_{ii}^- > 0, \\ 0, & \text{otherwise,} \end{cases} \qquad [V_0]_{ii} = \begin{cases} 1/\sqrt{D_{ii}^+}, & \text{if } D_{ii}^+ > 0, \\ 0, & \text{otherwise,} \end{cases} \tag{8}$$

where $D_{ii}$ denotes the degree of node $i$, as aforesaid.

In the case of undirected graphs, where the adjacency matrix $A$ is symmetric, each `JGE` layer admits a simplified variant, referred to as the Light Joint Graph Evolution (`LJGE`) layer. This formulation exploits the symmetry of $Q_t$ to reduce both computational overhead and the number of learnable parameters. The update equations for the `LJGE` are given by:

$$H_{t+1} = \Theta_t \, H_t^\top \, U_t \, Q_t \, U_t \, H_t,$$
$$Q_{t+1} = \sigma\big(H_{t+1} \, \Theta_t^\top\big), \tag{9}$$

where $\Theta_t \in \mathbb{R}^{r_{t+1} \times f}$ is the only learnable parameter matrix at layer $t$, and $\sigma$ denotes an elementwise activation function. By leveraging the symmetry of $Q_t$, this design yields a more lightweight and efficient alternative to the full `JGE` formulation.

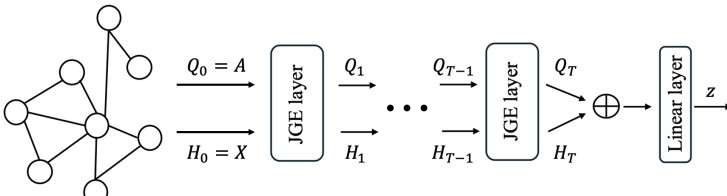

Figure 1: Node-level sparsification Pipeline. The operator $\oplus$ denotes the concatenation of the vectorized (flattened) forms of $Q_T$ and $H_T$.

## 2.2 Node Sparsification

`SpecNet` performs node pruning by leveraging the final representations $Q_T$ and $H_T$ produced by the last `JGE` layer (at step $T$). These matrices are first vectorized and concatenated into a single feature vector, which is then fed into a feedforward layer equipped with a Gumbel–sigmoid activation. The output is a binary selection mask $z \in \{0, 1\}^n$, where each entry $z_i$ indicates whether node $i$ is retained. The mask is transformed into a diagonal matrix $Z = \mathrm{diag}(z_1, \ldots, z_n)$, which is used to extract the subgraph induced by the selected nodes, with updated adjacency matrix $ZAZ$ and feature matrix $ZX$. In the case of directed graphs, post-processing may be necessary to eliminate isolated nodes resulting from the removal of both their in-neighbors and out-neighbors. The node-level sparsification approach is visually illustrated in Figure 1.

## 2.3 Spectral Concordance

The proposed loss function, termed as Spectral Concordance (`SC`) loss, measures the discrepancy between the leading spectra of the Laplacian of the original input graph and that of the graph synthesized by `SpecNet`. This choice is motivated by the fact that the eigenvalues of the graph Laplacian capture fundamental structural properties such as connectivity, clustering tendencies, and diffusion dynamics, as detailed in Appendix E.1.

The combinatorial Laplacian is most commonly defined in terms of the adjacency matrix $A \in \{0, 1\}^{n \times n}$ and degree matrix $D \in \mathbb{N}^{n \times n}$ as $L = D - A$. While this definition suffices for undirected graphs (where $A$ is symmetric), it does not generalize cleanly to directed graphs, which admit both an Out-Degree Laplacian $L^- = D^- - A$ and an In-Degree Laplacian $L^+ = D^+ - A$, that may be non-symmetrical.

An alternative formulation uses the incidence matrix $B \in \{-1, 0, 1\}^{n \times m}$, in which case $L = BB^\top$ provides a unified definition that applies equally to directed and undirected graphs. As shown in Appendix B, this incidence-based Laplacian is symmetric and positive semidefinite. Although valid, this formulation is computationally inefficient, as the incidence matrix $B \in \mathbb{R}^{n \times m}$ scales with the number of edges $m$, which can be comparable to $n^2$ in dense graphs. Fortunately, an equivalent and more compact representation for directed graphs is derived in Appendix C, hence the Laplacian matrix can be computed as:

$$L = BB^\top = \begin{cases} D - A, & \text{if the graph is undirected,} \\ D - (A + A^\top), & \text{otherwise}. \end{cases} \tag{10}$$

This identity enables Laplacian computation using only node-level structures, avoiding the explicit construction of the incidence matrix.

To ensure strict positive definiteness, the shifted Laplacian can be defined as:

$$L^* = L + \alpha_1 I, \tag{11}$$

with $\alpha_1 \in \mathbb{R}_{>0}$ and $I$ as the identity matrix. Appendix D proves that $L^*$ is nonsingular, symmetric, and positive definite, implying that all its eigenvalues are real and strictly positive.

Let $[\lambda_1 \geq \lambda_2 \geq \ldots]_{L^*}$ denote the eigenvalues of $L^*$ in descending order, and let Let $Z \in \{0, 1\}^{n \times n}$ be the diagonal selection matrix indicating the retained nodes as output of the `SpecNet`. The

spectral component of the SC loss function compares the top $k_1$ eigenvalues of the original and generated graphs:

$$\mathcal{L}_{\text{Laplace}}(L_A^*, L_{ZAZ}^*) = \frac{\left\| [\lambda_1, \ldots, \lambda_{k_1}]_{L_A^*} - [\lambda_1, \ldots, \lambda_{k_1}]_{L_{ZAZ}^*} \right\|_2}{\sum_{i,j:\, i \neq j}^n [L_A^*]_{ij}}, \tag{12}$$

where $L_A^*$ is the shifted Laplacian of the original adjacency matrix $A$, while $L_{ZAZ}^*$ is the shifted Laplacian of the generated adjacency matrix $ZAZ$. The summation term in the denominator is introduced to normalize the numerator of the loss function. This normalization is motivated by *Gerschgorin's Circle Theorem*, which provides bounds on the location of the eigenvalues of a matrix (Varga, 2004). In the specific case where the matrix is symmetric and positive definite, all eigenvalues are real and positive. This implies that the lower bound of the spectrum is zero. The use of the summation in the denominator thus ensures that the scale of the loss is properly adjusted, preventing unbounded growth due to large row sums (which influence the Gerschgorin discs), and guarantees numerical stability by keeping the loss within a meaningful range.

Beyond the spectral structure of the graph Laplacian, we also consider the alignment of the latent feature space induced by SpecNet. Specifically, we introduce an auxiliary term that penalizes spectral discrepancies between the input features $X \in \mathbb{R}^{n \times f}$ and the final feature representation $ZX \in \mathbb{R}^{r_T \times p_T}$.

Define the shifted Gram matrices:

$$M_X^* = X^\top X + \alpha_2 I, \qquad M_{ZX}^* = (ZX)^\top ZX + \alpha_2 I = X^\top ZX + \alpha_2 I, \tag{13}$$

where $\alpha_2 > 0$ ensures that both matrices are positive definite. Let $[\lambda_1 \geq \lambda_2 \geq \ldots]_{M_X^*}$ and $[\lambda_1 \geq \lambda_2 \geq \ldots]_{M_{ZX}^*}$ denote their ordered eigenvalues. A loss component, similar to Equation 12, comparing the top $k_2$ eigenvalues of $M_X^*$ and $M_{H_T}^*$, can be defined:

$$\mathcal{L}_{\text{Gram}}(M_X, M_{ZX}) = \frac{\left\| [\lambda_1, \ldots, \lambda_{k_2}]_{M_X^*} - [\lambda_1, \ldots, \lambda_{k_2}]_{M_{ZX}^*} \right\|_2}{\sum_{i,j:\, i \neq j}^n |[M_X^*]_{ij}|}. \tag{14}$$

where the denominator ensures, once again, the normalization by the Gershgorin radius. This term encourages the dominant modes of variation in the learned features to match those of the original input, and, as a consequence, it serves as a regularizer, promoting the preservation of global structure and expressivity in the learned feature space.

The Spectral Concordance (SC) is defined as a weighted combination of the Laplacian and Gram alignment losses introduced in Equations 12 and 14:

$$\mathcal{L}(L_A^*, M_X^*, L_{ZAZ}^*, M_{ZX}^*) = 1 - e^{-\mathcal{L}_{\text{Laplace}}(L_A^*, L_{ZAZ}^*)} + \beta \left( 1 - e^{-\mathcal{L}_{\text{Gram}}(M_X^*, M_{ZX}^*)} \right), \tag{15}$$

where $\beta > 0$ controls the trade-off between preserving the input graph's topology and retaining the feature structure, while the exponential functions contribute to bounding the loss terms in the range $(0, 1]$. In the specific setting of node-level sparsification, where the input and output graphs share the same dimensions, a regularization term is added to discourage trivial identity mappings:

$$\mathcal{L}_{\text{Spar}}(L_A^*, M_X^*, L_{ZAZ}^*, M_{ZX}^*) = \mathcal{L}(L_A^*, M_X^*, L_{ZAZ}^*, M_{ZX}^*) + \frac{\lambda}{n} \operatorname{tr}(Z), \tag{16}$$

where $\lambda > 0$ is a regularization coefficient that controls the degree of sparsification introduced by the network in the generated graph. The trace term, $\operatorname{tr}(Z) = \sum_i Z_{ii}$, penalizes the number of selected nodes, thereby promoting compact subgraph generations and reducing the risk of trivially replicating the input.

A deep and comprehensive discussion about the motivation, stability, and time and space complexity of SpecNet is provided in Appendix E.

## 3 EXPERIMENTAL TEST-BED AND RESULTS

We evaluate the proposed approach on five real-world attributed graphs: Cora, Citeseer, Actors, PubMed and Twitch-EN. A summary of their topological statistics is reported in Table 1, while the

Table 1: Real-world datasets.

| Dataset | Graph Type | Nodes | Edges | Attributes | Type |
|---|---|---|---|---|---|
| Cora | Citation | 2,708 | 5,429 | 1,433 | Directed |
| Citeseer | Citation | 3,312 | 4,591 | 3,703 | Directed |
| Actors | Co-occurrence | 7,600 | 29,926 | 932 | Directed |
| PubMed | Citation | 19,717 | 88,648 | 500 | Undirected |
| Twitch-EN | Social | 7,126 | 70,648 | 128 | Undirected |

descriptions and the links to access them are reported in Appendix F.

To validate the proposed method, we first conducted a graph-level analysis comparing the original graph with its sparsified version produced by our approach, focusing on quantitative measures. In this analysis, we considered two categories of metrics: *connection-based*, which capture both local properties such as node degree and global properties related to clustering or community structure, and *spectral-based*, derived from the eigenvalues and eigenvectors of the graph. The connection-based metrics include the size of the Largest Connected Component (LCC) $n_{LCC}$, the average node degree $\bar{k}$ as well as the average in-degree $\bar{k}_{in}$ and out-degree $\bar{k}_{out}$, and the modularity $M$. While the spectral measures include the Minimum Absolute Spectral Similarity (MASS) $\delta_{min}$ and the epidemic threshold $\tau_c$. The description of each metric is provided in Appendix G.

In a second set of experiments, we compared our method against existing sparsification techniques. Specifically, we considered: (i) Random Uniform Sparsifier (RUS), which randomly samples edges from the adjacency matrix to construct a sparsified graph; (ii) Spielman Sparsifier (SS) (Spielman & Srivastava, 2011), which relies on effective resistance values of edges for sparsification; (iii) the KSJ (Jaccard Similarity) and KSCT (Common Triangles) methods proposed in (Kim et al., 2022), which measure edge importance to guide sparsification; and (iv) D-Spar (Liu et al., 2023), a neural-based sparsification approach. Since our experiments cover both directed and undirected graphs, all compared methods were adapted to properly account for edge directionality. Further details on these approaches are provided in Appendix H.

**Results.** Table 2 shows how the structural and spectral properties of the graphs evolve after sparsification via `SpecNet`, under different reduction levels (i.e., number of preserved eigenvalues $\lambda$), reporting the mean and standard deviation over 10 runs. To show which topological traits are preserved or altered with respect to the original graph, we also report the reference metric values computed on the input graph (shown in the row immediately above each dataset's sparsified results).

After sparsifying with `SpecNet`, in Cora, the size of the largest connected component does not decrease monotonically with the number of retained eigenvalues. This is due to the non-monotonic number of edges preserved by the sparsification procedure: for higher numbers of eigenvalues (e.g., 32), more edges are selected compared to some intermediate cases, which allows additional nodes to remain connected or rejoin the LCC. The average degrees decrease proportionally with the reduction level, while the graph retains its modular structure, as evidenced by stable modularity scores. Also the MASS remains relatively high (above 0.65), approaching 0.85 for larger numbers of eigenvalues. This indicates that even after sparsification, the spectral structure of Cora is largely preserved. The epidemic threshold is preserved hence demonstrating that `SpecNet` keeps the network robustness level of the original graph.

Over Citeseer, `SpecNet` achieves effective sparsification while maintaining the core structure of the graph. The LCC size and the average degree are reduced as expected, but the size of the largest connected component decreases as the number of the eigenvalues increases. However, the main connected component still contains a significant portion of nodes. Modularity remains relatively unchanged, suggesting that the community structure is preserved. Accordingly, the MASS stays above 0.71, showing that the sparsified graphs retain a substantial part of the original spectral characteristics, with only moderate deviation. Also on this citation network the functional robustness of the sparsified graph remains stable.

Also on Actors, despite the sparsification inducted, the modularity remains stable, indicating that community structures are largely preserved. The MASS values are consistently high (above 0.91 for the smallest eigenvalue counts), showing that the spectral properties of the network are well maintained. The epidemic threshold is again preserved showing that the sparsification process does not significantly affect the network's key dynamical properties.

Table 2: `SpecNet` graph quantitative measures computed for different numbers of eigenvalues.

| Dataset | # of $\lambda$ | $n_{edges}$ | $n_{LCC}$ | $k$ | $k_{in}$ | $k_{out}$ | $M$ | $\delta_{min}$ | $\tau_c$ |
|---|---|---|---|---|---|---|---|---|---|
| Cora | - | 5,429 | 2,485 | 4.01 | 2.00 | 2.00 | 0.82 | - | 0.07 |
| | 2 | 3,599 ± 60 | 1,810 ± 33 | 2.66 ± 0.05 | 1.33 ± 0.02 | 1.33 ± 0.02 | 0.82 ± 0.01 | 0.65 ± 0.09 | 0.07 ± 0.00 |
| | 4 | 3,645 ± 67 | 1,828 ± 15 | 2.69 ± 0.05 | 1.35 ± 0.03 | 1.35 ± 0.03 | 0.82 ± 0.01 | 0.75 ± 0.10 | 0.07 ± 0.00 |
| | 8 | 3,465 ± 79 | 1,745 ± 30 | 2.56 ± 0.06 | 1.28 ± 0.03 | 1.28 ± 0.03 | 0.81 ± 0.01 | 0.80 ± 0.07 | 0.07 ± 0.00 |
| | 16 | 3,067 ± 39 | 1,559 ± 22 | 2.27 ± 0.03 | 1.13 ± 0.01 | 1.13 ± 0.01 | 0.80 ± 0.01 | 0.80 ± 0.08 | 0.07 ± 0.00 |
| | 32 | 3,551 ± 44 | 1,753 ± 25 | 2.62 ± 0.03 | 1.31 ± 0.02 | 1.31 ± 0.02 | 0.81 ± 0.01 | 0.85 ± 0.06 | 0.07 ± 0.00 |
| Citeseer | - | 4,591 | 2,110 | 2.77 | 1.39 | 1.39 | 0.89 | - | 0.07 |
| | 2 | 3,396 ± 59 | 1,499 ± 64 | 2.05 ± 0.04 | 1.03 ± 0.02 | 1.03 ± 0.02 | 0.89 ± 0.00 | 0.71 ± 0.11 | 0.07 ± 0.00 |
| | 4 | 3,357 ± 52 | 1,482 ± 64 | 2.03 ± 0.03 | 1.01 ± 0.02 | 1.01 ± 0.02 | 0.89 ± 0.00 | 0.73 ± 0.11 | 0.07 ± 0.00 |
| | 8 | 3,181 ± 55 | 1,388 ± 61 | 1.92 ± 0.03 | 0.96 ± 0.02 | 0.96 ± 0.02 | 0.88 ± 0.00 | 0.76 ± 0.11 | 0.07 ± 0.00 |
| | 16 | 2,812 ± 49 | 1,192 ± 44 | 1.70 ± 0.03 | 0.85 ± 0.02 | 0.85 ± 0.02 | 0.87 ± 0.00 | 0.76 ± 0.10 | 0.07 ± 0.00 |
| | 32 | 2,216 ± 32 | 908 ± 38 | 1.34 ± 0.02 | 0.67 ± 0.01 | 0.67 ± 0.01 | 0.86 ± 0.01 | 0.75 ± 0.10 | 0.07 ± 0.00 |
| Actors | - | 29,926 | 7,600 | 7.88 | 3.94 | 3.94 | 0.51 | - | 0.03 |
| | 2 | 18,583 ± 979 | 5,299 ± 164 | 4.89 ± 0.26 | 2.45 ± 0.13 | 2.45 ± 0.13 | 0.52 ± 0.01 | 0.91 ± 0.01 | 0.03 ± 0.00 |
| | 4 | 16,614 ± 5,877 | 4,548 ± 1,404 | 4.37 ± 1.55 | 2.19 ± 0.77 | 2.19 ± 0.77 | 0.50 ± 0.01 | 0.92 ± 0.01 | 0.03 ± 0.00 |
| | 8 | 20,814 ± 1,294 | 5,756 ± 265 | 5.48 ± 0.34 | 2.74 ± 0.17 | 2.74 ± 0.17 | 0.52 ± 0.01 | 0.93 ± 0.01 | 0.03 ± 0.00 |
| | 16 | 20,085 ± 371 | 5,641 ± 68 | 5.29 ± 0.10 | 2.64 ± 0.05 | 2.64 ± 0.05 | 0.52 ± 0.00 | 0.94 ± 0.01 | 0.03 ± 0.00 |
| | 32 | 20,323 ± 227 | 5,738 ± 44 | 5.35 ± 0.06 | 2.67 ± 0.03 | 2.67 ± 0.03 | 0.53 ± 0.00 | 0.94 ± 0.01 | 0.03 ± 0.00 |
| PubMed | - | 44,324 | 19,717 | 4.50 | 4.50 | 4.50 | 0.77 | - | 0.04 |
| | 2 | 21,629 ± 734 | 10,812 ± 133 | 2.19 ± 0.07 | 2.19 ± 0.07 | 2.19 ± 0.07 | 0.78 ± 0.00 | 0.40 ± 0.13 | 0.05 ± 0.00 |
| | 4 | 23,478 ± 668 | 11,109 ± 119 | 2.38 ± 0.07 | 2.38 ± 0.07 | 2.38 ± 0.07 | 0.77 ± 0.01 | 0.44 ± 0.11 | 0.05 ± 0.00 |
| | 8 | 30,823 ± 3,256 | 13,933 ± 1,490 | 3.13 ± 0.33 | 3.13 ± 0.33 | 3.13 ± 0.33 | 0.76 ± 0.01 | 0.52 ± 0.14 | 0.05 ± 0.00 |
| | 16 | 29,086 ± 2,800 | 13,677 ± 1,139 | 2.95 ± 0.28 | 2.95 ± 0.28 | 2.95 ± 0.28 | 0.77 ± 0.01 | 0.51 ± 0.14 | 0.05 ± 0.00 |
| | 32 | 22,844 ± 844 | 11,239 ± 331 | 2.32 ± 0.09 | 2.32 ± 0.09 | 2.32 ± 0.09 | 0.78 ± 0.01 | 0.48 ± 0.20 | 0.05 ± 0.00 |
| Twitch-EN | - | 35,324 | 7,126 | 9.91 | 9.91 | 9.91 | 0.45 | - | 0.02 |
| | 2 | 24,790 ± 526 | 5,013 ± 84 | 6.96 ± 0.15 | 6.96 ± 0.15 | 6.96 ± 0.15 | 0.44 ± 0.01 | 0.73 ± 0.15 | 0.02 ± 0.00 |
| | 4 | 23,906 ± 721 | 4,604 ± 138 | 6.71 ± 0.20 | 6.71 ± 0.20 | 6.71 ± 0.20 | 0.44 ± 0.01 | 0.74 ± 0.14 | 0.02 ± 0.00 |
| | 8 | 25,768 ± 3,368 | 4,838 ± 777 | 7.23 ± 0.95 | 7.23 ± 0.95 | 7.23 ± 0.95 | 0.43 ± 0.01 | 0.78 ± 0.10 | 0.02 ± 0.00 |
| | 16 | 25,659 ± 4,675 | 4,846 ± 1,078 | 7.20 ± 1.31 | 7.20 ± 1.31 | 7.20 ± 1.31 | 0.44 ± 0.01 | 0.85 ± 0.07 | 0.02 ± 0.00 |
| | 32 | 24,915 ± 2,443 | 4,519 ± 633 | 6.99 ± 0.69 | 6.99 ± 0.69 | 6.99 ± 0.69 | 0.44 ± 0.01 | 0.86 ± 0.03 | 0.02 ± 0.00 |

In PubMed, the LCC size decreases proportionally with the reduction in the number of edges, while the average degree similarly decreases. Modularity remains again stable across sparsification–levels, MASS values, however, are lower compared to the other smaller datasets. This is expected given the large size and density of the graph: sparsification with few retained eigenvalues removes a substantial fraction of edges, inducing more pronounced deviations in the spectral structure, and thus a lower minimum abstract spectral similarity. The epidemic threshold shows a minor increase: this minor change, typical when sparsifying large networks Kuga & Tanimoto (2022), is due to a small reduction in the largest eigenvalue of the adjacency matrix, reflecting a minimal loss in the network's diffusion capacity. Overall, the sparsification preserves the robustness of the network.

For Twitch-EN, the LCC size and average degree both decrease as expected with stronger sparsification. The network maintains a relatively low modularity but consistent with the original, reflecting its weak community structure. MASS values remain above 0.73, indicating that the main spectral characteristics are preserved. Finally, also for this dataset, the epidemic threshold remains stable.

The results in Table 3 demonstrate that `SpecNet` consistently achieves high MASS values across all datasets, particularly on Cora, Citeseer, and Pubmed, effectively preserving the original spectral structure compared to locally-based methods (KSJ, KSCT) and D-Spar, which show much lower values in many cases. This indicates that the reduction performed by `SpecNet` maintains the global properties of the graph, which is critical for tasks such as community detection or information propagation. Compared to RUS, `SpecNet` is more stable, especially on datasets like Actors where random edge selection leads to higher variance, while Spielman Sparsifier (SS) performs well as expected for a spectral method, yet `SpecNet` is often competitive or superior, particularly at medium-to-high values of $\lambda$ (8–32), highlighting the effectiveness of its spectral regularization component. Local attribute-based variants such as KSJ and KSCT generally achieve lower MASS on datasets like Citeseer and Pubmed, indicating that purely local methods struggle to preserve the global characteristics of large graphs, whereas `SpecNet` maintains consistent values thanks to its joint transformation of topology and node features. D-Spar shows very low MASS values on Cora and Citeseer, demonstrating that, while useful for GNN preprocessing, it does not preserve the global structure of the sparsified graphs, unlike `SpecNet`, which produces graphs that remain faithful to the original. Finally, `SpecNet` maintains relatively high MASS even for small numbers of eigenvalues ($\lambda$ = 2–4), showing that a good global representation can be retained with few spectral dimensions, while larger values of $\lambda$ (16–32) result in stable or improved performance, confirming the model's ability to leverage additional spectral information without introducing noise.

## 4 RELATED WORK

Our contributions address two complementary aspects of graph learning: (i) the design of a novel neural layer that jointly embeds node features and structural information, and (ii) a loss function for spectral sparsification that removes nodes while preserving global properties. We therefore organize the related work into two groups: methods for *graph embeddings and joint structure–feature learning*, and approaches to *graph sparsification*.

Table 3: Comparison with other state-of-the-art sparsification methods in terms of MASS. For all the sparsifiers, the number of network links that are kept, i.e., the sparsification threshold, is the same adopted by our sparsifier.

| Dataset | # of $\lambda$ | RUS | SS | KSJ | KSCT | D-SPAR | SpecNet |
|---|---|---|---|---|---|---|---|
| Cora | 2 | $0.55 \pm 0.03$ | $0.65 \pm 0.00$ | $0.64 \pm 0.01$ | $0.54 \pm 0.01$ | $0.18 \pm 0.00$ | $0.65 \pm 0.09$ |
| | 4 | $0.55 \pm 0.04$ | $0.75 \pm 0.00$ | $0.73 \pm 0.01$ | $0.65 \pm 0.02$ | $0.18 \pm 0.00$ | $0.75 \pm 0.10$ |
| | 8 | $0.54 \pm 0.03$ | $0.76 \pm 0.01$ | $0.72 \pm 0.00$ | $0.64 \pm 0.01$ | $0.18 \pm 0.00$ | $0.80 \pm 0.07$ |
| | 16 | $0.45 \pm 0.04$ | $0.78 \pm 0.01$ | $0.74 \pm 0.01$ | $0.67 \pm 0.02$ | $0.20 \pm 0.00$ | $0.80 \pm 0.08$ |
| | 32 | $0.56 \pm 0.04$ | $0.83 \pm 0.00$ | $0.80 \pm 0.01$ | $0.63 \pm 0.01$ | $0.18 \pm 0.00$ | $0.85 \pm 0.06$ |
| Citeseer | 2 | $0.42 \pm 0.03$ | $0.61 \pm 0.00$ | $0.52 \pm 0.00$ | $0.52 \pm 0.00$ | $0.21 \pm 0.00$ | $0.71 \pm 0.11$ |
| | 4 | $0.41 \pm 0.02$ | $0.61 \pm 0.01$ | $0.52 \pm 0.00$ | $0.52 \pm 0.00$ | $0.22 \pm 0.01$ | $0.73 \pm 0.11$ |
| | 8 | $0.40 \pm 0.03$ | $0.61 \pm 0.01$ | $0.52 \pm 0.00$ | $0.52 \pm 0.00$ | $0.21 \pm 0.00$ | $0.76 \pm 0.11$ |
| | 16 | $0.39 \pm 0.05$ | $0.61 \pm 0.01$ | $0.52 \pm 0.00$ | $0.52 \pm 0.00$ | $0.18 \pm 0.01$ | $0.76 \pm 0.10$ |
| | 32 | $0.39 \pm 0.05$ | $0.61 \pm 0.01$ | $0.52 \pm 0.00$ | $0.52 \pm 0.00$ | $0.21 \pm 0.00$ | $0.75 \pm 0.10$ |
| Actors | 2 | $0.63 \pm 0.04$ | $0.82 \pm 0.00$ | $0.41 \pm 0.00$ | $0.83 \pm 0.00$ | $0.73 \pm 0.00$ | $0.91 \pm 0.01$ |
| | 4 | $0.59 \pm 0.21$ | $0.58 \pm 0.47$ | $0.47 \pm 0.00$ | $0.83 \pm 0.03$ | $0.71 \pm 0.02$ | $0.92 \pm 0.01$ |
| | 8 | $0.71 \pm 0.04$ | $0.82 \pm 0.00$ | $0.41 \pm 0.00$ | $0.86 \pm 0.00$ | $0.73 \pm 0.01$ | $0.93 \pm 0.01$ |
| | 16 | $0.67 \pm 0.02$ | $0.59 \pm 0.47$ | $0.47 \pm 0.45$ | $0.57 \pm 0.47$ | $0.68 \pm 0.15$ | $0.94 \pm 0.01$ |
| | 32 | $0.68 \pm 0.02$ | $0.82 \pm 0.00$ | $0.58 \pm 0.22$ | $0.88 \pm 0.00$ | $0.76 \pm 0.02$ | $0.94 \pm 0.01$ |
| Pubmed | 2 | $0.40 \pm 0.08$ | $0.40 \pm 0.00$ | $0.40 \pm 0.10$ | $0.38 \pm 0.13$ | $0.02 \pm 0.02$ | $0.40 \pm 0.13$ |
| | 4 | $0.40 \pm 0.01$ | $0.41 \pm 0.09$ | $0.41 \pm 0.00$ | $0.41 \pm 0.02$ | $0.02 \pm 0.02$ | $0.44 \pm 0.11$ |
| | 8 | $0.49 \pm 0.06$ | $0.46 \pm 0.00$ | $0.48 \pm 0.04$ | $0.47 \pm 0.08$ | $0.07 \pm 0.07$ | $0.52 \pm 0.14$ |
| | 16 | $0.44 \pm 0.02$ | $0.48 \pm 0.15$ | $0.41 \pm 0.03$ | $0.41 \pm 0.03$ | $0.04 \pm 0.04$ | $0.51 \pm 0.14$ |
| | 32 | $0.38 \pm 0.06$ | $0.42 \pm 0.01$ | $0.41 \pm 0.19$ | $0.41 \pm 0.10$ | $0.02 \pm 0.02$ | $0.48 \pm 0.20$ |
| Twitch-EN | 2 | $0.61 \pm 0.08$ | $0.70 \pm 0.20$ | $0.02 \pm 0.00$ | $0.33 \pm 0.08$ | $0.35 \pm 0.07$ | $0.73 \pm 0.15$ |
| | 4 | $0.58 \pm 0.02$ | $0.62 \pm 0.00$ | $0.01 \pm 0.00$ | $0.34 \pm 0.00$ | $0.34 \pm 0.00$ | $0.74 \pm 0.14$ |
| | 8 | $0.58 \pm 0.05$ | $0.72 \pm 0.14$ | $0.01 \pm 0.00$ | $0.34 \pm 0.04$ | $0.35 \pm 0.02$ | $0.78 \pm 0.10$ |
| | 16 | $0.55 \pm 0.02$ | $0.72 \pm 0.00$ | $0.01 \pm 0.00$ | $0.33 \pm 0.03$ | $0.34 \pm 0.00$ | $0.85 \pm 0.07$ |
| | 32 | $0.60 \pm 0.07$ | $0.72 \pm 0.16$ | $0.01 \pm 0.00$ | $0.34 \pm 0.06$ | $0.35 \pm 0.05$ | $0.86 \pm 0.03$ |

## 4.1 GRAPH EMBEDDINGS AND JOINT STRUCTURE–FEATURE LEARNING

Learning expressive node embeddings has been a cornerstone of graph representation learning. Early unsupervised models such as DeepWalk (Perozzi et al., 2014) and node2vec (Grover & Leskovec, 2016) rely on random walks to capture local connectivity patterns, but they neglect node attributes and provide no control over graph structure. Spectral clustering (von Luxburg, 2007) similarly embeds nodes in eigenspaces of the Laplacian, but operates on fixed graphs and lacks feature integration.

Message-passing neural networks, including GCN (Kipf & Welling, 2017), GraphSAGE (Hamilton et al., 2017), and GAT (Veličković et al., 2018), combine structural neighborhoods with node features through aggregation schemes. These models enable inductive learning and leverage both topology and attributes, but they assume static input graphs and suffer from oversmoothing in deeper layers (Li et al., 2018; Oono & Suzuki, 2020). Moreover, structure and features are typically entangled into a single embedding space, limiting flexibility. Extensions such as DropEdge (Rong et al., 2020) or attention-weight pruning (Veličković et al., 2018) introduce heuristic sparsification, but without principled guarantees.

Our proposed layer departs from these approaches by *jointly learning embeddings and structural transformations*. Through bilinear reparameterizations of the Laplacian, it synthesizes adaptive graph topologies that are not restricted to subgraphs of the input. This allows the model to discover intermediate structures aligned with both node features and spectral properties, providing richer and more flexible embeddings than static or purely feature-agnostic methods.

## 4.2 GRAPH SPARSIFICATION

Graph reduction techniques can be broadly divided into sparsification, coarsening, and condensation (Hashemi et al., 2024). We focus on sparsification, which seeks sparse graphs that approximate the original structure while reducing complexity.

**Classical and spectral methods.** Benczúr & Karger (1996) introduced cut-preserving sparsifiers, while Spielman & Srivastava (2011) and Batson et al. (2013) developed nearly-linear algorithms sampling edges according to effective resistance. These approaches preserve Laplacian spectra and commute times with strong guarantees, but rely on costly pseudoinverses and do not scale easily. Extensions address weighted, directed, and dynamic graphs (Kapralov et al., 2014), yet remain detached from learning objectives.

**Heuristic and geometric pruning.** Simpler approaches remove weak or redundant edges by weight thresholding (Yan et al., 2018), neighborhood similarity (Satuluri et al., 2011), or community-preserving heuristics (Leskovec et al., 2009). Backbone extraction methods such as Noise-Corrected

filtering (Coscia & Neffke, 2017; Coscia & Rossi, 2019) retain statistically significant edges, while Ricci curvature (Zhang et al., 2024) or walk-based pruning (Razin et al., 2023) exploit local geometry or stochastic connectivity. These methods are efficient but heuristic, offering no formal control over spectral preservation.

**Neural sparsification.** Recent models integrate sparsification into learning pipelines. NeuralSparse (Zheng et al., 2020) learns edge scores for supervised tasks, but outputs strict subgraphs tied to labels. GSGAN (Wu & Chen, 2020) uses adversarial training to preserve communities via random walks, while GraphSAINT (Zeng et al., 2020) samples subgraphs for mini-batch training. PRI (Yu et al., 2022) matches Laplacian spectra through Jensen–Shannon divergence, but fixes graph size and requires large matrices. DSpar (Liu et al., 2023) approximates effective resistance by node degrees to accelerate training. While effective, these models rely on supervision, heuristics, or restricted formulations.

**Our sparsification.** In contrast, our approach formulates sparsification as a *spectral alignment problem* with feature integration. A Laplacian-based loss preserves global spectral properties, a Gram-matrix loss enforces feature geometry alignment, and a trace penalty provides explicit sparsity control. This differentiable formulation enables node-level pruning within end-to-end training, offering a general and unsupervised alternative to heuristic, task-specific, or structure-only methods.

Unlike prior work in the literature, we are able to provide both the adjacency matrix and the feature matrix in a way that remains consistent with the intrinsic properties of the nodes. The only exception occurs when the features are purely structural, in which case they can be recomputed from the reduced adjacency matrix.

## 5    CONCLUSION AND FUTURE WORK

We introduced Spectral Preservation Network (`SpecNet`), a novel neural architecture that stacks Joint Graph Evolution (`JGE`) layers to jointly evolve both a graph structure and its node representations. The model is equipped with a new loss function, Spectral Concordance (`SC`), which enables principled node-level sparsification by aligning structural and feature spectra. By reparameterizing the graph Laplacian, `SpecNet` preserves global properties while overcoming the rigidity of static message passing that characterizes the existing graph neural network literature. Empirically, our method outperforms current state-of-the-art approaches on standard benchmarks, particularly under the MASS metric, demonstrating the effectiveness of spectrum-driven sparsification.

This work opens several promising directions for future research. First, beyond node pruning, the `JGE` layer naturally supports *graph condensation*: rather than selecting subsets of the original graph, it can synthesize entirely new graphs and feature matrices that retain the information content of the input data. Second, extending the formulation beyond square adjacency matrices would allow `JGE` to operate on heterogeneous relational data, where multiple groups of objects (potentially belonging to different domains and containing varying numbers of elements) interact through non-square incidence patterns. Such a generalization would substantially broaden the applicability of our framework to domains ranging from multi-relational networks to cross-modal representation learning.

## REPRODUCIBILITY STATEMENT

To ensure reproducibility of our results, we provide both theoretical and experimental support. Intuitions, motivation, formal proofs of the main theorems and additional derivations are included in the appendix, which clarify the assumptions, the applicability domain, and the limitations of the proposed model. For the experimental validation, we release the full implementation of our method, together with the preprocessing pipeline and training scripts, available at `https://anonymous.4open.science/r/CA43`. These materials allow independent researchers to reproduce the reported results and explore further applications of our approach.

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

# A  STABILITY OF THE JOINT GRAPH EVOLUTION LAYER

**Theorem 1.** *Let $Q \in \mathbb{R}^{r \times s}$ be any real matrix. Define its row– and column–absolute sums by*

$$u_i = \sum_{j=1}^{s} |Q_{ij}|, \quad v_j = \sum_{i=1}^{r} |Q_{ij}|.$$

*Form the diagonal scaling matrices $U \in \mathbb{R}^{r \times r}$ and $V \in \mathbb{R}^{s \times s}$ via*

$$U_{ii} = \begin{cases} 1/\sqrt{u_i}, & u_i > 0, \\ 0, & u_i = 0, \end{cases} \qquad V_{jj} = \begin{cases} 1/\sqrt{v_j}, & v_j > 0, \\ 0, & v_j = 0. \end{cases}$$

*Then the normalized matrix*

$$\widehat{Q} = U Q V$$

*satisfies*

$$\|\widehat{Q}\|_{\mathrm{op}} \leq 1,$$

*i.e. $\widehat{Q}$ is non-expansive in the Euclidean norm, indicating by $\|\widehat{Q}\|_{\mathrm{op}}$ the induced spectral operator norm, i.e., the square root of the largest eigenvalue of $\widehat{Q}^{\top} \widehat{Q}$.*

*Proof.* Recall that for any matrix $M$ the induced spectral operator norm is:

$$\|M\|_{\mathrm{op}} := \sup_{x \neq 0} \frac{\|Mx\|_2}{\|x\|_2} = \sup_{\|x\|_2 = 1} \|Mx\|_2,$$

where $\|x\|_2 = (\sum_j x_j^2)^{1/2}$ is the Euclidean norm. It suffices to show $\|\widehat{Q}x\|_2 \leq 1$ for all unit vectors $x \in \mathbb{R}^s$.

Let $y = Vx$, then:

$$\widehat{Q}x = U(Qy).$$

Hence:

$$\|\widehat{Q}x\|_2^2 = \sum_{i=1}^{r} U_{ii}^2 \left( \sum_{j=1}^{s} Q_{ij} y_j \right)^2.$$

Since $U_{ii}^2 = 1/u_i$ when $u_i > 0$ and zero otherwise,

$$\|\widehat{Q}x\|_2^2 = \sum_{i:u_i > 0} \frac{1}{u_i} \left( \sum_{j=1}^{s} Q_{ij} y_j \right)^2.$$

Bounding each summand using the triangle inequality followed by Cauchy–Schwarz:

$$\left| \sum_{j=1}^{s} Q_{ij} y_j \right| \leq \sum_{j=1}^{s} |Q_{ij}| \, |y_j| = \sum_{j=1}^{s} \sqrt{|Q_{ij}|} \cdot \sqrt{|Q_{ij}|} \, |y_j|.$$

Thus, applying Cauchy–Schwarz on nonnegative vectors:

$$\left( \sum_{j=1}^{s} Q_{ij} y_j \right)^2 \le u_i \sum_{j=1}^{s} |Q_{ij}| \, y_j^2.$$

and thus:

$$\|\widehat{Q} \, x\|_2^2 \; \le \; \sum_{i=1}^{r} \sum_{j=1}^{s} |Q_{ij}| \, y_j^2 \; = \; \sum_{j=1}^{s} \left( \sum_{i=1}^{r} |Q_{ij}| \right) y_j^2 \; = \; \sum_{j=1}^{s} v_j \, y_j^2.$$

Finally, since $y_j = x_j / \sqrt{v_j}$ whenever $v_j > 0$ (and $x_j = y_j = 0$ if $v_j = 0$):

$$\sum_{j=1}^{s} v_j \, y_j^2 = \sum_{j=1}^{s} x_j^2 = \|x\|_2^2 = 1.$$

Hence $\|\widehat{Q} \, x\|_2^2 \le 1$ for all unit $x$, and taking the supremum yields $\|\widehat{Q}\|_{\mathrm{op}} \le 1$, as claimed. $\qquad\square$

## B    PROPERTIES OF THE LAPLACIAN MATRIX

**Theorem 2.** *Let $B \in \mathbb{R}^{n \times m}$ be any real matrix. Define:*

$$L = BB^T \; \in \; \mathbb{R}^{n \times n},$$

*then $L$ is symmetric and positive semidefinite.*

*Proof.* First:

$$L^T = (BB^T)^T = BB^T = L,$$

so $L$ is symmetric. Next, for any $x \in \mathbb{R}^n$, set $y = B^T x \in \mathbb{R}^m$. Then

$$x^T L \, x = x^T (BB^T) x = (B^T x)^T (B^T x) = y^T y = \sum_{k=1}^{m} y_k^2 \; \ge \; 0.$$

Hence $L$ is positive semidefinite. $\qquad\square$

## C    LAPLACIAN MATRIX FOR DIRECTED GRAPHS

**Theorem 3.** *Let $G = (V, E)$ be a directed graph on $n$ nodes (without self-loops), with adjacency matrix $A \in \{0, 1\}^{n \times n}$. Define a signed incidence matrix*

$$B \in \{-1, 0, 1\}^{n \times m},$$

*where $m = |E|$, by choosing an arbitrary but fixed orientation of each edge $e_k$ and setting*

$$B_{i,k} = \begin{cases} -1, & \text{if node } i \text{ is the tail of edge } e_k, \\ +1, & \text{if node } i \text{ is the head of edge } e_k, \\ 0, & \text{otherwise.} \end{cases}$$

*Let $D \in \mathbb{N}^{n \times n}$ be the diagonal matrix whose $i$th entry $D_{ii}$ equals the total degree of node $i$, i.e. the sum of its in- and out-degrees. Then*

$$B B^\top \; = \; D \; - \; (A + A^\top).$$

*That is, for general (asymmetric) $A$, the incidence-based Laplacian recovers the symmetrized combinatorial Laplacian.*

*Proof.* We verify the equality entry-wise.

**Diagonal entries ($i = j$).**

$$\left[ BB^\top \right]_{ii} = \sum_{k=1}^{m} B_{i,k}^2 = \sum_{k : \, i \in e_k} 1 = D_{ii}.$$

**Off-diagonal entries** ($i \neq j$).

$$\left[BB^\top\right]_{ij} = \sum_{k=1}^{m} B_{i,k} B_{j,k}.$$

A nonzero contribution arises only when $e_k$ connects $i$ and $j$. If $e_k$ is oriented $i \to j$, then $B_{i,k} = -1$, $B_{j,k} = +1$, so $B_{i,k}B_{j,k} = -1$. If $e_k$ is oriented $j \to i$, then $B_{i,k} = +1$, $B_{j,k} = -1$, again $B_{i,k}B_{j,k} = -1$. Hence

$$\sum_{k=1}^{m} B_{i,k} \, B_{j,k} = -\big(\delta\{i \to j \in E\} + \delta\{j \to i \in E\}\big) = -(A_{ij} + A_{ji}).$$

Where $\delta\{\cdot\}$ is a binary function whose value is 1 if its argument is `True`, 0 otherwise. That is, $\left[B\,B^\top\right]_{ij} = -\big(A + A^\top\big)_{ij}$.

Combining diagonal and off-diagonal cases yields

$$B\,B^\top = D \,-\, (A + A^\top),$$

as claimed. $\qquad\square$

## D  ON THE SINGULARITY OF $L + \alpha I$

**Theorem 4.** *Let $L \in \mathbb{R}^{n \times n}$ be a symmetric and positive semidefinite matrix, and let $\alpha \in \mathbb{R}_{>0}$ be a positive scalar that is not an eigenvalue of $L$. Then $L + \alpha I$ is symmetric, positive definite, and therefore nonsingular.*

*Proof.* Since both $L$ and the identity matrix $I$ are symmetric, their sum $L + \alpha I$ is symmetric as well. To prove that $L + \alpha I$ is positive definite, consider any nonzero vector $x \in \mathbb{R}^n$. Then,

$$x^T(L + \alpha I)x = x^T L x + \alpha x^T x.$$

Because $L$ is positive semidefinite, $x^T L x \geq 0$. Moreover, since $\alpha > 0$ and $x \neq 0$, we have $\alpha x^T x > 0$. Thus, $x^T(L + \alpha I)x > 0$ for all $x \neq 0$, and hence $L + \alpha I$ is positive definite. Positive definite matrices are invertible, so $L + \alpha I$ is nonsingular. $\qquad\square$

## E  INTUITION AND ANALYSIS OF THE SPECTRAL PRESERVATION NETWORK

### E.1  MOTIVATION

Spectral sparsification Batson et al. (2013) has emerged as a principled approach for reducing the density of large graphs while preserving their global structural and dynamical properties. Unlike heuristic or naive pruning strategies scoring all edges/nodes uniformly and pruning them based on a prefixed sparsity level Chen et al. (2023) considering the lowest weights or local topological criteria (e.g., low node degree or triangle count Liu et al. (2023)), spectral sparsification explicitly preserves the global spectral geometry of the graph, that is maintaining the essential eigenstructure of the graph's Laplacian matrix, which encodes rich information about the global topology, connectivity, and dynamics of the network Chung (1997); von Luxburg (2007). While weight-based thresholding may eliminate edges that appear weak or redundant, it provides no formal guarantees about the impact on connectivity, diffusion processes, or the spectrum of the Laplacian. In contrast, spectral sparsification methods construct subgraphs that maintain critical algebraic and dynamical properties of the original graph within a well-defined approximation bound.

Specifically, a graph $G'$ is said to be an $\varepsilon$-spectral sparsifier of a graph $G$ if the quadratic form of the Laplacians satisfies $(1 - \varepsilon)x^T L x \leq x^T L' x \leq (1 + \varepsilon)x^T L x$ for all vectors $x \in \mathbb{R}^n$, where $L$ and $L'$ denote the Laplacian matrices of $G$ and $G'$, respectively. This condition ensures that key properties such as *effective resistance, commute times, and spectral clustering behavior are approximately maintained in the sparsified representation*. In particular, the *effective resistance* Klein & Randić (1993) between nodes, which quantifies the influence of an edge on global connectivity, plays a central role in modeling diffusion and current flow through the network. Maintaining

approximate effective resistances guarantees that edge importance in terms of global communication is preserved. Similarly, *commute times*, defined as the expected number of steps a random walker takes to travel from one node to another and return, are tightly linked to the spectrum of the Laplacian and to resistance distances. These metrics reflect how efficiently information or influence spreads in the network. Furthermore, preserving the Laplacian spectrum also retains the embedding space used in *spectral clustering* Ding et al. (2024), where the eigenvectors of the Laplacian encode low-dimensional representations that capture community structure, modularity, or functional subsystems. As a result, spectral sparsification allows the reduced graph to faithfully approximate the original graph's geometry and signal propagation behavior, which is essential in applications such as brain network analysis, semi-supervised learning, and the design of graph neural network filters.

### E.2 On Exact Binary Optimization

An alternative approach to node-level sparsification would be to solve the combinatorial problem

$$\min_{Z \in \{0,1\}^n} \mathcal{L}(L_A^*, M_X^*, L_{ZAZ}^*, M_{ZX}^*) + \lambda \operatorname{tr}(Z). \tag{17}$$

However, this formulation entails a combinatorial search over $2^n$ binary masks, making it intractable even for moderately sized graphs. Instead, our method leverages a continuous relaxation of $Z$ via Gumbel-sigmoid sampling, enabling efficient gradient-based optimization. This allows for scalable training while still encouraging discrete sparsification through the trace penalty. Additionally, the use of spectral alignment losses ensures a balanced trade-off between structural and feature preservation.

### E.3 Computational Complexity and Stability

To assess the theoretical and practical feasibility of the proposed Spectral Preservation Network, an analysis of its stability, space complexity, and time complexity is presented.

#### E.3.1 Model Stability

As detailed in Appendix A, the normalization of the structural matrix $Q_t$ via diagonal matrices $U_t$ and $V_t$ ensures that the transformation $U_t Q_t V_t$ remains non-expansive with respect to the Euclidean norm, satisfying $\|U_t Q_t V_t\|_2 \leq 1$. This property constrains the Lipschitz constant of each JGE layer, mitigating risks of feature explosion or vanishing across multiple layers.

The non-expansiveness contributes to enhanced numerical stability and consistent gradient propagation, which in turn supports more reliable convergence during optimization. These benefits are particularly relevant in deep graph architectures, where instabilities are commonly encountered.

Additional stability is provided by the use of shifted Laplacian and Gram matrices (Equations 11 and 13), whose eigenvalues are strictly positive, as demonstrated in Appendix D. This guarantees that the transformations remain well-conditioned, avoiding numerical issues associated with near-singular matrices.

Collectively, these mechanisms promote robustness to input perturbations and enable stable end-to-end training of deep graph networks.

#### E.3.2 Space Complexity

Each JGE layer introduces a temporary tensor $J_{t+1} \in \mathbb{R}^{p_t \times p_t}$ and three learnable parameter matrices: $\Theta_t \in \mathbb{R}^{r_{t+1} \times p_t}$, $\Phi_t \in \mathbb{R}^{p_t \times r_{t+1}}$, and $\Psi_t \in \mathbb{R}^{p_t \times p_{t+1}}$, for every $t \in \{1, \ldots, T\}$. The size of the learnable parameters remains both tractable and explicitly controllable, as their dimensions are specified by design and are independent of the size or structure of the input graph. The only exception is the first layer, where $p_0 = f$ depends on the dimensionality of the input features. In typical applications, however, $f$ is significantly smaller than the number of nodes $n$ or edges $m$, making this dependency negligible. In cases where $f$ is unusually large, standard dimensionality reduction techniques, such as Principal Component Analysis (PCA), can be applied to the input feature matrix $X$ during preprocessing.

Assuming constant dimensions across layers, i.e., $r_t = r$ and $p_t = p$ for all $t$, the total space required by the `SpecNet` model is given by:

$$\mathcal{O}\big(T\, p\, (p+r)\big). \tag{18}$$

In the node-level sparsification setting, an additional feedforward layer processes a concatenation of the flattened matrices $Q_T$ and $H_T$, producing an output vector of size $n$. This results in an overall space complexity of:

$$\mathcal{O}\big(T\, p\, (p+r) + r\, (r+f)\, n\big). \tag{19}$$

This accounts for both model parameters and the additional memory required by the final selection mechanism.

### E.3.3  TIME COMPLEXITY

**Forward Pass.**  To analyze the time complexity of the `SpecNet` architecture, the operations within each `JGE` layer, as defined in Equation 4, are examined in detail. Let $r_t = r$ and $p_t = p$ for all layers $t \in \{1, \ldots, T\}$, as is typically assumed for simplicity.

Each layer involves the following steps:

- Construction of diagonal normalization matrices $U_t, V_t \in \mathbb{R}^{r \times r}$ from $Q_t \in \mathbb{R}^{r \times r}$, requiring $\mathcal{O}(r^2)$.
- Elementwise normalization to compute $Q'_t = U_t Q_t V_t$, which adds another $\mathcal{O}(r^2)$ (as $U_t$ and $V_t$ are diagonal).
- Bilinear projection $Q''_t = H_t^\top Q'_t H_t$, resulting in a matrix in $\mathbb{R}^{p \times p}$ and costing $\mathcal{O}(pr^2 + rp^2)$.
- Computation of the intermediate tensor $J_{t+1} = \Theta_t Q''_t \in \mathbb{R}^{r \times p}$, which requires $\mathcal{O}(rp^2)$.
- Final updates of $Q_{t+1} \in \mathbb{R}^{r \times r}$ and $H_{t+1} \in \mathbb{R}^{r \times p}$ through nonlinear transformations, both costing $\mathcal{O}(pr^2)$.

Summing the dominant terms, the per-layer cost is $\mathcal{O}(pr^2 + rp^2)$, therefore, the total time complexity of the forward pass through a `SpecNet` network with $T$ `JGE` layers is:

$$\mathcal{O}\big(T\, (pr^2 + rp^2)\big). \tag{20}$$

This estimate represents the worst-case scenario. In practice, the use of optimized GPU matrix libraries can reduce the empirical cost significantly via parallelization and memory-efficient algorithms, often achieving sub-cubic runtime behavior.

In the case of node-level graph sparsification, a final projection to the original node space is required, introducing an additional cost of $\mathcal{O}(r\, (r+p)\, n)$. The overall forward complexity then becomes:

$$\mathcal{O}\big(T\, (pr^2 + rp^2) + r\, (r+p)\, n\big). \tag{21}$$

**Loss Function Complexity.**  The computation of the shifted Laplacian matrix $L^*_{ZAZ} \in \mathbb{R}^{n \times n}$ (Equations 2.3, 10, and 11) depends on the type of graph:

- *Undirected graphs*: computing $L = D - ZAZ$ costs $\mathcal{O}(n^2)$, as $D$ is diagonal and $Z$ is diagonal and binary.
- *Directed graphs*: computing $L = D - (ZAZ + (ZAZ)^\top)$ incurs $\mathcal{O}(n^2)$ as well.

The shifted Laplacian, by adding the scalar shift $\alpha_1 I$, costs $\mathcal{O}(n)$, thus, in the worst case (directed setting), its computation $\mathcal{O}(n^2)$ time. In contrast, the shifted Gram matrix $M^*_{ZX} \in \mathbb{R}^{f \times f}$ (Equation 13) is formed from $X^\top Z X + \alpha_2 I$, which has cost $\mathcal{O}(nf^2)$.

The cost of computing all eigenvalues of a dense matrix in $\mathbb{R}^{n \times n}$ is typically $\mathcal{O}(n^3)$ Golub & van Loan (2013). However, when the matrix is symmetric and positive definite, as in the case of this work, efficient algorithms exist:

- In the dense setting, the *MRRR algorithm* (Multiple Relatively Robust Representations) can reduce the cost to $\mathcal{O}(n^2)$ under favorable conditions Dhillon et al. (2006).

- In the sparse setting, *iterative methods* such as the *Lanczos algorithm* Cullum & Willoughby (2002) compute the top-$k$ eigenvalues and corresponding eigenvectors with cost $\mathcal{O}(k \cdot \texttt{nnz})$, where $\texttt{nnz}$ is the number of non-zero entries.

Computing the norm of the difference of the two sets of eigenvalues costs $\mathcal{O}(k)$, assuming $k_1 = k_2 = k$, that is negligible.

Overall, the asymptotical worst-case upper bound for computing the full Spectral Concordance loss function is:

$$\mathcal{O}(\min(n^2, k \cdot \texttt{nnz}) + nf^2). \tag{22}$$

In the node-level sparsification setting, the trace regularization term (Equation 16) adds a negligible $\mathcal{O}(n)$.

**Summary.** Let $s$ denote the number of training epochs. Table 4 summarizes the overall time complexity for both training and inference.

| Phase | Time Complexity |
|---|---|
| Training | $\mathcal{O}\big(s\,(T\,(pr^2 + rp^2) + r\,(r+p)\,n + \min(n^2, k \cdot \texttt{nnz}) + nf^2)))$ |
| Inference | $\mathcal{O}(T\,(pr^2 + rp^2) + r\,(r+p)\,n)$ |

Table 4: Time complexity of the `SpecNet` architecture in its two principal configurations, for both training and inference.

## F  DATA

In our experimental assessement, we used the following datasets:

- **Cora**[1] is a citation network where nodes represent scientific publications and edges denote citation links, i.e., a citation from a publication to another. Node features are bag-of-words vectors built from a dictionary of unique terms, with binary indicators for word presence.

- **Citeseer**[2] is another citation graph of research papers. As in Cora, nodes correspond to publications and edges to citation links, with bag-of-words feature vectors.

- **Actors**[3] is a directed co-occurrence graph in which nodes represent actors and directed edges indicate that one actor is mentioned in the Wikipedia page of another. Node features are bag-of-words representations of the corresponding page content.

- **PubMed**[4] is a large-scale citation graph where nodes are scientific articles and edges represent citation relationships, treated as undirected. Node attributes are TF-IDF vectors extracted from textual content.

- **Twitch-EN**[5] is a social network where each node corresponds to a Twitch user and edges represent mutual follow relationships. Node features encode user-level metadata. The dataset contains overlapping communities and densely connected subgroups.

All graphs are pre-processed by removing self-loops and duplicate edges.

---

[1] `https://linqs.org/datasets/#cora`
[2] `https://github.com/ZPowerZ/citeseer-dataset/tree/master`, `https://linqs.org/datasets/#citeseer-doc-classification`
[3] `https://pytorch-geometric.readthedocs.io/en/2.6.0/generated/torch_geometric.datasets.Actor.html#torch_geometric.datasets.Actor`
[4] `https://pytorch-geometric.readthedocs.io/en/2.6.0/generated/torch_geometric.datasets.CitationFull.html#torch_geometric.datasets.CitationFull`
[5] `https://pytorch-geometric.readthedocs.io/en/2.6.0/generated/torch_geometric.datasets.Twitch.html#torch_geometric.datasets.Twitch`

# G   EVALUATION METRICS

**Connection-based metrics.**   Connection-based metrics capture both local connectivity and global network behavior through community-level structure. We consider three metrics: (i) the size of the Largest Connected Component (LCC) $n_{LCC}$, (ii) the average node degree $\bar{k}$, and (iii) the modularity $M$.

The size of the largest connected component $n_{LCC}$ measures the number of nodes in the largest connected subgraph in $G$. Tracking the LCC provides a straightforward estimate of how many nodes remain part of the principal connected structure.

The degree of a node $i$ is defined as $k_i = \sum_{j \neq i} a_{ij}$, where $a_{ij}$ denotes the adjacency matrix entry of the graph $G$. This metric corresponds to the number of neighbors of a node. In our analysis, we focus on the average node degree $\bar{k}$, which measures the mean number of neighbors per node and provides a concise measure of the network's overall connectivity. For directed graphs, we also consider the average in-degree $\bar{k}_{in}$, i.e., the mean number of incoming edges, and the average out-degree $\bar{k}_{out}$, i.e., the mean number of outgoing edges.

The modularity $M$ quantifies the extent to which a network is organized into densely connected clusters of nodes, with relatively few connections between different clusters. To assess each subject's community modularity, in our analysis, we first identify the communities within the networks by using the *Louvain* algorithm Blondel et al. (2008). Once the communities are detected, the modularity $M$ of the partitioning is computed as:

$$M = \frac{1}{2m} \sum_{ij} \left( \omega_{ij} - \frac{k_i k_j}{2m} \right) \delta(c_i, c_j) \tag{23}$$

where $m$ is the sum of the edge weights of $G$, $w_{ij}$ is the weight of edge $(i, j)$ in $G$, $k_i$ and $k_j$ are the weighted degrees of nodes $i$ and $j$ respectively, $c_i$ and $c_j$ are the communities of the corresponding nodes, and $\delta$ is the Kronecker function which yields 1 if $i$ and $j$ are in the same community, that is $c_i = c_j$, zero otherwise. Networks with high modularity are characterized by strong intra-community connectivity and weak inter-community connectivity. If the modularity of the input graph and the modularity of the sparsified graph remain similar, the sparsification preserves the community structure, meaning the sparsified graph retains key intra-community edges, removing edges likely belonging to inter-community connections, which are less critical for modularity.

Since sparsification inherently reduces the number of nodes/edges, both the size of the largest connected component and the average node degree decrease accordingly. These measures are therefore not used to assess structural preservation, but rather to provide an estimate of the reduction rate in terms of node connectivity. In contrast, metrics such as modularity are employed to evaluate the extent to which the community structure is preserved after sparsification.

**Spectral-based metrics.**   Spectral measures derive from the eigenvalues and eigenvectors of graph matrices. We consider three metrics: (i) the Minimum Absolute Spectral Similarity (MASS) $\delta_{min}$ and (ii) the epidemic threshold $\tau_c$.

The Minimum Absolute Spectral Similarity Yan et al. (2018) $\delta_{min}$ is a quality index measuring the difference between the spectral properties of a graph and its sparsified version after edge removals. The measure specifically quantifies the difference between the Laplacian $L$ of $G$ and the Laplacian $L'$ of the sparsifier $G'$. The minimum relative spectral similarity (MRSS) between $L'$ and $L$ is usually computed as:

$$\delta_{min}^R = min_{\forall z} \frac{z^T L' z}{z^T L z} \tag{24}$$

where $z$ can be any vector with $N$ elements and $z^T L' z$ is the Laplacian quadratic form. The vector $z$ intuitively represents the direction along which the difference between the two graphs is measured. As such, the minimum value of similarity reflects the worst case. However, if $G'$ disconnects into components, the MRSS value becomes zero, making the use of this measure unstable for many optimization algorithms. An alternative viable measure is the *absolute spectral similarity* proposed

in Yan et al. (2018):

$$\delta(z) = 1 - \frac{z^T \Delta L z}{z^T [\lambda_1]_L z} \tag{25}$$

where $[\lambda_1 \geq \lambda_2 \geq \ldots]_L$ are the eigenvalues of $L$, and $\Delta L = \Delta D - \Delta A$ is the Laplacian of the difference graph $\Delta G$ having the same set of nodes of $G$ and the set of edges removed during the sparsification. Since the input vector $z$ is variable, considering the worst-case scenario, the *minimum absolute spectral similarity* (MASS) is

$$\delta_{min} = min_{|z|=1}\left(1 - \frac{z^T \Delta L z}{[\lambda_1]_L}\right) = 1 - \frac{[\lambda_1]_{\Delta L}}{[\lambda_1]_L} \tag{26}$$

where $[\lambda_1 \geq \lambda_2 \geq \ldots]_{\Delta L}$ are the eigenvalues of the difference Laplacian $\Delta L$ and, without loss of generality, only the unit length vectors $|z| = 1$ are considered.

The MASS is able to practically quantify the robustness of a network at a mesoscopic (i.e., communities) level when edges are removed and the network disconnects. Ranging between 0 and 1, the MASS offers a practical and computationally efficient similarity measure between the original graph and its version after the edge reduction, indicating whether the spectral properties of the original graph are kept or not after its perturbation.

The epidemic threshold $\tau_c$: the largest eigenvalue of the adjacency matrix of $G$ also known as *spectral radius* and denoted with $\lambda_1$, is considered a powerful character of dynamic processes on complex networks since it characterizes the spread of viruses and synchronization processes Li et al. (2011) Van Mieghem et al. (2009). It is a common practice to choose the inverse of the spectral radius, the *epidemic threshold* $\tau_c$ as a measure for *robustness*: the larger the epidemic threshold, the more robust a network is against the spread of a virus. In epidemiology theory, the inverse of $\lambda_1$, in fact, characterizes the threshold of a phase transition Castellano & Pastor-Satorras (2010) over which the network shifts from a virus-free state with zero infected nodes to fractions of infected nodes where the virus is persistent. The epidemic threshold formula

$$\tau_c = \frac{1}{\lambda_1} \tag{27}$$

is rigorously demonstrated in the N-intertwined approximation, named NIMFA, of the exact SIS (Susceptible-Infected-Susceptible) model Van Mieghem et al. (2009). The spectral radius which is computed in $O(m)$, and hence the epidemic threshold, is strictly related to the path capacity of the network. In Restrepo et al. (2007), it is demonstrated that $\lambda_1$ can be approximated by $N3/N2$, where $N_k$ is the total number of walks in $k$ hops. Van Mieghem et al. proved that $N3/N2$ is a lower bound for the spectral radius Van Mieghem et al. (2010). If the sparsified graph has a similar epidemic threshold, the sparsification preserves the network's robustness and also its ability to transmit information or infections, retaining key high-degree and central edges. The epidemic threshold thus serves as an indicator of both network robustness and *information preservation*, reflecting the network's ability to maintain connectivity and support effective information propagation despite sparsification.

## H CONTESTANT METHODS

SpecNet is compared by considering the following contestant methods:

- Random Uniform Sparsifier (RUS): randomly samples edges from a given adjacency matrix $A$ to create a sparsified graph. The sparsification is performed uniformly, meaning each edge is equally likely to be selected, regardless of its weight or structural role. The approach is simple and unbiased, but may discard important edges.

- Spielman Sparsifier (SS): spectral sparsification through the effective resistance values of the edges. Based on the foundational work by Spielman and Srivastava (Spielman & Srivastava, 2011), the approach retain edges with higher effective resistance $\omega_{ij}$ computed as

$$\omega_{ij} = l_{ii}^+ + l_{jj}^+ - 2l_{ij}^+, \tag{28}$$

where $l_{ij}^+$ are the elements of the Moore-Penrose *pseudoinverse* matrix $L^+$ of the weighted Laplacian matrix of $G$.

- Kim et al. (Kim et al., 2022) edge attribute based sparsification (KS): a class of methods assigning edge importance based on topological features computed locally for each edge. Edges are then sparsified by selecting those with the highest attribute-based scores, enhancing local structure preservation. Specifically, three variants are considered:

  - KSJ (Jaccard Similarity): edge weight is computed as the Jaccard index between the neighborhoods of its two endpoints $i$ and $j$:

  $$\mathrm{J}(i,j) = \frac{|N(i) \cap N(j)|}{|N(i) \cup N(j)|}. \tag{29}$$

  - KSCT (Common Triangles): edge weight is proportional to the number of triangles that include the edge, promoting edges involved in tightly connected clusters:

  $$\mathrm{T}(i,j) = |N(i) \cap N(j)| - 2. \tag{30}$$

- D-Spar Liu et al. (2023): prepares a smaller graph for a GNN (e.g., GCN, GraphSAGE, GAT, etc.) to train or infer on. D-Spar indirectly affects the GNN by deciding what structure the GNN will see and learn from. More specifically, this preprocessing strategy computes a score for each edge as

  $$\mathrm{Dscore}(i,j) = \frac{1}{D_{ii}} + \frac{1}{D_{jj}}, \tag{31}$$

  where $D_{ii}$ and $D_{jj}$ are the degrees of nodes $i$ and $j$ respectively. Then, a percentage of edges with the highest scores are kept, while all the other edges are removed. This scoring scheme prioritizes edges connecting low-degree nodes, which are typically more crucial for maintaining the global structure of sparse graphs.

Since our experiments involve both directed and undirected graphs, the compared sparsification methods were adapted accordingly to handle directionality. Edge-based scores like Jaccard similarity and common triangles were computed using both in- and out-neighbors, and triangle counts considered directed motifs such as cycles and feedforward structures. Finally, degree-based quantities were computed by distinguishing in-degree and out-degree of each node (i.e., $D_{ii}^+$, $D_{ii}^-$).

