# OpenReview forum: "Spectral Neural Graph Sparsification"
_ICLR.cc/2026/Conference — ICLR 2026 Conference Withdrawn Submission_

### Official Review · Reviewer_bM5E · 2025-10-28

**Soundness:** 1
**Presentation:** 1
**Contribution:** 1
**Rating:** 0
**Confidence:** 5

**Summary:**

The paper proposes a spectral neural architecture (SpecNet) that jointly learns graph structure and node features via a “Joint Graph Evolution” layer and then performs node-level sparsification guided by a spectral alignment loss, with the goal of producing a smaller graph that preserves the original graph’s Laplacian spectrum and feature geometry.

**Strengths:**

1. The paper attempts to couple structure learning and sparsification: it proposes a Joint Graph Evolution (JGE) layer that is intended to update both the graph connectivity ($Q_{t+1}$) and node features ($H_{t+1}$), together with a Spectral Concordance loss meant to preserve spectral properties (e.g., Laplacian behavior, global structure) after pruning nodes.

2. The experiments at least try to evaluate sparsification not only by edge count but also by higher-level structural indicators (largest connected component size, modularity, epidemic threshold, MASS)

**Weaknesses:**

The paper would benefit from a substantial revision under the guidance of a more senior researcher. In its current form, it’s difficult to evaluate. The abstract and introduction together span only about one page, are very high-level, and begin describing the method almost immediately. There is no sufficiently developed background, related work section or problem formulation, and the motivation is unclear. As a result, it’s not clear what specific task the paper is targeting or what the concrete contributions are.

-------

The presentation is not professional. The paper spends nearly a full page of the main text re-explaining basic graph preliminaries such as adjacency matrices, incidence matrices, and in-/out-degree. While these concepts are relevant, they are undergraduate-level material and could be summarized more concisely or moved to the appendix. At the same time, Tables 1–3 are dense, visually cramped, and placed in layouts with large unused whitespace, which makes them difficult to read.

-------

In Section 2.1, the proposed Joint Graph Evolution (JGE) layer is defined to output updated graph structure $Q_{t+1} \in \mathbb{R}^{r_{t+1} \times r_{t+1}}$ and node features $H_{t+1} \in \mathbb{R}^{r_{t+1} \times p_{t+1}}$.
   However, the intermediate matrix $J_{t+1}$ is stated to be in $\mathbb{R}^{p_t \times p_t}$, and then the paper defines
   $Q_{t+1} = \sigma_1(J_{t+1} \Phi_t)$ with $\Phi_t \in \mathbb{R}^{p_t \times r_{t+1}}$.
   Multiplying a $(p_t \times p_t)$ matrix by a $(p_t \times r_{t+1})$ matrix gives shape $(p_t \times r_{t+1})$, not $(r_{t+1} \times r_{t+1})$ as claimed for $Q_{t+1}$.
   The same mismatch appears again for $H_{t+1}$.  This is a basic correctness issue in the model specification.

-------

The paper repeatedly refers to $Q_{t+1}$ as “a transformed graph topology with updated edge weights and a redefined node set,” implying that it is a learned adjacency for the next layer. But there is no constraint that $Q_{t+1}$ be a valid adjacency-like operator (e.g., nonnegative, sparse, symmetric, or even well-conditioned). It is simply $Q_{t+1} = \sigma_1(J_{t+1}\Phi_t)$, i.e. an arbitrary dense learned matrix. The paper also claims stability by normalizing $Q_t$ via diagonal scalings $U_t$ and $V_t$ so that $U_t Q_t V_t$ is “non-expansive,” but that argument only applies to the intermediate product, not to the final $Q_{t+1}$ after additional learned projections and nonlinearities. In other words, neither the “this is a new adjacency” claim nor the “this is stable” claim is actually enforced by the formulation given.

---------

Only a limited set of baselines is considered, and they are largely standard sparsifiers or older heuristics. Authors should also consider modern deep learning-based baselines.

**Questions:**

N/A.

---

### Official Review · Reviewer_9M5h · 2025-10-31

**Soundness:** 2
**Presentation:** 3
**Contribution:** 2
**Rating:** 2
**Confidence:** 3

**Summary:**

The paper attempts to evolve the graph during training adaptively and also sparsify it to address the reliance on a fixed graph structure and oversmoothing. The paper proposes a new architecture, the Joint Graph Evolution layer (JGE layer), and a spectral concordance loss to sparsify the graph. Empirical results on robust structural and spectral properties after sparsification and a high MASS value, which indicates that the method maintains the global properties of a graph.

**Strengths:**

1. The paper explores the idea of evolving the graph structure during training, a direction that has been relatively underexplored in existing research, highlighting its novel contribution.
2. The proposed method achieves superior performance compared to established baselines, demonstrating its effectiveness and supporting the originality of the approach.

**Weaknesses:**

1. The paper would benefit from additional comparisons with related works. For instance, graph pooling methods also dynamically compress graph structures for improved representation, often employing similar equations. A more apparent distinction and comparison with such approaches would strengthen the contribution.
2. Some key experiments, particularly an ablation study on the loss functions, are missing and should be included to clarify the contribution of each component.
3. Even though the authors stated that fixed graph structure can fall into scalability and expressive issues, empirical analysis of whether the proposed method mitigates these issues is underexplored.
4. The time complexity, which grows quadratically with the number of nodes, raises concerns about the scalability of the approach to large-scale graphs.

**Questions:**

- Could the authors provide a more detailed explanation of how the right-hand side of Equation (13) is derived?
- Can $Z$ be extended to explain the crucial edges in a given input graph?
- When conventional GNN methods are optimized using the reduced graph, do they maintain high performance when evaluated on the original graph?

---

### Official Review · Reviewer_Nrsr · 2025-10-31

**Soundness:** 3
**Presentation:** 3
**Contribution:** 3
**Rating:** 4
**Confidence:** 3

**Summary:**

SpecNet introduces a spectral-preserving neural sparsifier that combines a Joint Graph Evolution (JGE) layer with a Spectral Concordance (SC) loss. The JGE layer co-evolves adjacency and feature matrices through bilinear transformations, while the SC loss aligns the leading eigenvalues of Laplacian and Gram matrices to maintain spectral and feature consistency. It performs node-level sparsification preserving both structure and attributes.

**Strengths:**

1. Novel joint architecture that evolves structure and features instead of treating graphs as static.
2. Spectral-theoretic objective preserves global connectivity and dynamical properties.
3. Extensive evaluation across 5 datasets and multiple sparsity ratios

**Weaknesses:**

1. Experiments primarily assess spectral metrics (MASS, τc) rather than downstream performance.
2. Eigen-decomposition can be computationally costly for large graphs

**Questions:**

1. How does SpecNet affect downstream GNN accuracy after sparsification?
2. What are the guidelines for choosing k₁/k₂/β/λ? (Optional)
3. Can JGE handle heterogeneous or non-square incidence graphs?
4. How scalable is the eigenvalue computation in high-dimensional settings? (Main issue)

---

### Official Review · Reviewer_NBhJ · 2025-11-02

**Soundness:** 1
**Presentation:** 1
**Contribution:** 1
**Rating:** 0
**Confidence:** 4

**Summary:**

This work introduces the Spectral Preservation Network (SpecNet), which consists of a joint graph evolution layer to mitigate oversmoothing by adaptively reshaping the topology, and a spectral concordance loss that sparsifies the graph to remove uninformative nodes.

**Strengths:**

1. Graph sparsification is indeed a practical problem when dealing with large graphs.
2. The writing of the proposal is easy to follow, which also applies to most parts of the paper.

**Weaknesses:**

1. The motivation is unclear and lacks real-world support. For instance, in line 42, the authors mention “evolve the graph.” However, if the graph encoder is already changing continuously, the extracted information from the graph would also change without needing to modify the graph structure itself. What is the deeper reason for evolving the graph structure?
2. Dropping nodes may limit the applicability of the proposed GNN. For example, link prediction would no longer be fully feasible.
3. Could you elaborate further on the design of the proposed single layer in Equation (4)? The intuition behind it is unclear.
4. What are the empirical results regarding the improvement of oversmoothing? Additionally, no evaluation metrics are provided.

**Questions:**

See the weaknesses listed above.

---

### Note · Authors · 2025-11-24

I have read and agree with the venue's withdrawal policy on behalf of myself and my co-authors.